# An Effective Parameter Analysis for Sending-or-Not-Sending Quantum Key Distribution with Untrusted Light Sources

**DOI:** 10.3390/e27060547

**Published:** 2025-05-22

**Authors:** Jiajian Huang, Weigang Li, Yucheng Qiao

**Affiliations:** 1Guangxi Key Laboratory of Cryptography and Infomation Security, Guilin University of Electronic Technology, Guilin 541004, China; jiajianhuang@mails.guet.edu.cn; 2School of Computer Science and Information Security, Guilin University of Electronic Technology, Guilin 541004, China; lwg1997@mails.guet.edu.cn

**Keywords:** quantum key distribution, twin fields, sending-or-not-sending, light source monitoring

## Abstract

The twin-field (TF) protocol is a key protocol in quantum key distribution (QKD) that enables remote key distribution, achieving a maximum secure transmission distance of over 500 km. However, the TF protocol still faces several security issues in real-world environments. To address the issue of untrusted sources, one effective solution is to introduce a light-source monitoring module into the system. Analysis shows that a solution based on untagged bits (UBs) can achieve ideal monitoring performance. This solution can capture UB signals to accurately estimate key parameters in the protocol’s security analysis, ultimately deriving a tight bound for the secure bit rate. Simulations show that this solution approximates the performance of ideal light sources in the presence of untrusted sources and effectively mitigates the impact of light-source fluctuations. It outperforms other solutions in key performance metrics, such as transmission distance.

## 1. Introduction

Quantum key distribution (QKD) is a secure communication protocol with strong security guarantees. In 1984, the BB84 protocol [1] emerged, marking the beginning of quantum communication. Since then, various protocols have emerged, including the BBM92 protocol [2], SARG04 protocol [3], differential phase protocol [4], and six-state protocol [5]. On the other hand, rapid theoretical advancements in QKD have been accompanied by significant experimental achievements. In 1992, Bennett and Brassard successfully built the first QKD experimental system [6] and demonstrated the key distribution process of the BB84 protocol. In 1999, Bechmann-Pasquinucci and Tittel proposed the use of high-dimensional quantum qubits [7] for QKD to enhance its information-carrying capacity, replacing traditional two-dimensional qubits. In 2002, Cerf and Bourennane analyzed the security of d-level QKD systems [8]. The feasibility of high-dimensional QKD has been gradually demonstrated by researchers with studies focusing on transmission distance [9,10] and security [11,12,13]. Although the communication distance and key length of this system are limited, it proved that QKD is feasible. This indicates that QKD research is continuously evolving and becoming more sophisticated. However, as research progresses, scholars have identified several security vulnerabilities in QKD applications. These include the photon number-splitting (PNS) attack [14], the fake state (FS) attack [15], the time-shift (TS) attack [16], and the detector-blinding (DB) attack [17].

Scholars have proposed various methods and protocols to counter the above attacks. In 2004, Gottesman, Lo, N. Lütkenhaus, and Preskill proposed an analytical method for assessing the security of the BB84 protocol with non-ideal devices, which was known as the GLLP [18] theory. This theory effectively extends the security of the BB84 protocol to practical applications, analyzing the real-world security of QKD. Subsequent theoretical advancements, such as the decoy state method [19] and optimized parameter estimation techniques, have been developed to improve the performance of practical systems [20,21,22], enhancing both security and efficiency. To address side-channel attack threats faced by detectors, scholars have proposed the measurement-device-independent (MDI) protocol [23].

To improve the performance of practical QKD systems, Lucamarini et al. proposed the twin-field (TF) protocol [24] in 2018. Due to its significant advantages in transmission, it has sparked widespread research. Many subsequent enhanced protocols, referred to as TF-type protocols, further improved both the security and performance of the system. Among the TF-type protocols, sending-or-not-sending (SNS) protocol [25] is more widely used. The SNS protocol achieves long-distance secure transmission by configuring a series of sending or not sending signals. Recently, researchers have analyzed the security of the SNS protocol under different situations [26,27,28,29,30]. Researchers also have proposed various methods to enhance the performance of the SNS protocol. For example, by using the independent lasers [31], introducing the actively odd parity pairing (AOPP) method [32,33,34] and applying the phase postselection [35], the transmission range of the SNS protocol can be significantly extended. Nonetheless, practical implementation challenges persist. Researchers have addressed the practical security issues of the SNS protocol, particularly those related to the light source [36,37,38,39,40,41,42,43,44]. In our preliminary work, we investigated light source monitoring. Although this work is theoretically significant, it is not practical for real-world system implementation. In this paper, we propose a practical solution, untagged bits (UBs) [45], to effectively address the light source security challenges in SNS protocol.

This paper is organized as follows. In Section 2, we describe the steps of the SNS protocol, analyze its security, and estimate key parameters. In Section 3, we present numerical simulations and analyze the results. Finally, Section 4 concludes the paper.

## 2. SNS Protocol with UB

### 2.1. Introduction of SNS Protocol

Although the structure of the SNS protocol is similar to that of the TF protocol, it optimizes the signal preparation and post-processing steps. The steps of the SNS protocol are as follows [25]:At each time window *i*, Alice (Bob) determines whether it is a signal or decoy window. If it is a decoy window, Alice (Bob) prepares a coherent state $|\sqrt{\mu_{d}}e^{i\delta +i\gamma }\rangle$ and sends it to Charlie. If it is a signal window, Alice (Bob) prepares a coherent state $|\sqrt{\mu_{s}}e^{i\delta +i\gamma }\rangle$ with probability ϵ and sends it to Charlie. $\mu_{d}$ and $\mu_{s}$ represent the photon intensity of the decoy state and the signal state, $\delta \in \left\{\delta_{A},\delta_{B}\right\}$ represents the random phase, and $\gamma \in \{\gamma_{A},\gamma_{B}\}$ represents the phase offset of the channel.Charlie receives the states sent by Alice and Bob and publishes all the measurement results for the effective events. Effective events are defined as follows: (1) When Alice and Bob simultaneously decide on the signal window, Alice (Bob) decides to send the signal, while Bob (Alice) decides not to send it, corresponding to Charlie announcing only detector $D_{0(1)}$ clicks. (2) When Alice and Bob simultaneously decide on the decoy window, they prepare coherent states with the same intensity, and in this window, the random phases $\delta_{A}$ and $\delta_{B}$ satisfy [25](1)$$1-|\cos(\delta_{A}-\delta_{B})|⩽|\lambda |,$$
corresponding to Charlie announcing only detector $D_{1(0)}$ clicks. The value of λ is determined by the size of the phase slice chosen by Alice and Bob, as described in Ref. [24].After Charlie publishes all the measurement results, Alice and Bob announce all the windows and the details of the decoy windows to classify the data accordingly and determine the parameters of the security key formula.The secret key rate of the SNS protocol has been given as [25,26](2)$$R=2\epsilon \left(1-\epsilon \right)P_{1}^{L}\left(\mu_{s}\right)s_{1}^{L}\left[1-H\left(e_{1}^{U}\right)\right]-fS_{Z}H\left(E_{Z}\right).$$

ϵ represents the probability that Alice (Bob) sends the signal state to Charlie during the signal window. $P_{1}^{L}\left(\mu_{s}\right)$ refers to the lower bound of the probability of sending a signal that contains a single photon. We define *Z*-data [26] as events in which Alice and Bob simultaneously select the signal window. $Z_{1}$-data refers to effective events where one party chooses to send the signal while the other does not. $s_{1}^{L}\;\mathrm{and}\;e_{1}^{U}$ represent the lower bound of the count rate and the upper bound of the phase error rate of single-photon events in the $Z_{1}$-data. $S_{Z}$ and $E_{Z}$ represent the count rate and bit error rate of the signal, *f* is the error correction efficiency, and $H\left(x\right)=-xlog_{2}x-\left(1-x\right)log_{2}\left(1-x\right)$ refers to the binary Shannon entropy function.

Under asymptotic conditions, Equation (Equation 2) remains valid and can accurately estimate the parameters $s_{1}^{L}\;\mathrm{and}\;e_{1}^{U}$ [25].(3)$$s_{1}\geq s_{1}^{L}=\frac{p_{2}^{L}\left(\mu_{2}\right)\left[S_{\mu_{1}}-p_{0}^{U}\left(\mu_{1}\right)S_{\mu_{0}}\right]-p_{2}^{U}\left(\mu_{1}\right)\left[S_{\mu_{2}}-p_{0}^{L}\left(\mu_{2}\right)S_{\mu_{0}}\right]}{p_{2}^{U}\left(\mu_{2}\right)p_{1}^{U}\left(\mu_{1}\right)-p_{2}^{L}\left(\mu_{1}\right)p_{1}^{L}\left(\mu_{2}\right)},$$(4)$$e_{1}\leq e_{1}^{U}=\frac{S_{\mu_{1}}E_{\mu_{1}}-p_{0}^{L}\left(\mu_{1}\right)S_{\mu_{0}}E_{\mu_{0}}}{p_{1}^{L}\left(\mu_{1}\right)s_{1}^{L}}.$$

$\mu_{0}=0<\mu_{1}<\mu_{2}$ represents the three intensities of the decoy windows. $p_{0}^{L(U)}\left(\mu_{i}\right),p_{1}^{L(U)}\left(\mu_{i}\right),\mathrm{and}\;p_{2}^{L(U)}\left(\mu_{i}\right)\left(i=\left(0,1,2\right)\right)$ can be directly calculated as(5)$$p_{0}^{L(U)}\left(\mu_{i}\right)=P_{0,A}^{L(U)}\left(\mu_{i}\right)P_{0,A}^{L(U)}\left(\mu_{i}\right),$$(6)$$p_{1}^{L(U)}\left(\mu_{i}\right)=P_{0,A}^{L(U)}\left(\mu_{i}\right)P_{1,B}^{L(U)}\left(\mu_{i}\right)+P_{1,A}^{L(U)}\left(\mu_{i}\right)P_{0,B}^{L(U)}\left(\mu_{i}\right),$$(7)$$p_{2}^{L(U)}\left(\mu_{i}\right)=P_{0,A}^{L(U)}\left(\mu_{i}\right)P_{2,B}^{L(U)}\left(\mu_{i}\right)+P_{2,A}^{L(U)}\left(\mu_{i}\right)P_{0,B}^{L(U)}\left(\mu_{i}\right)+P_{1,A}^{L(U)}\left(\mu_{i}\right)P_{1,B}^{L(U)}\left(\mu_{i}\right).$$

### 2.2. Security Analysis

In the SNS protocol, Z-data are generated from effective events when Alice (Bob) sends a signal state while Bob (Alice) does not. Therefore, the security of the SNS protocol is equivalent to that of the BB84 protocol with a decoy-state scheme. Equation (Equation 2) holds under asymptotic conditions. However, the key challenge lies in accurately estimating $s_{1}^{L},e_{1}^{U}$. For X-data, when both communicating parties select decoy states of the same intensity, the output two-mode quantum state is given by $\left|\varphi \rangle =\right|\sqrt{\mu }e^{i\delta_{A}}\rangle \;⊗\;|\sqrt{\mu }e^{i\delta_{B}}\rangle$, where μ represents the average photon number of the decoy state. Although the random phases $\delta_{A}$ and $\delta_{B}$ in X-data satisfy Equation (Equation 1), indicating that $\delta_{A}-\delta_{B}$ is not random, the value of $\delta_{A}+\delta_{B}$ remains random. We define a new variable $\delta_{\pm }=\delta_{A}\pm \delta_{B}$. Under this new variable, the output quantum state becomes [25](8)$$\left|\varphi \rangle =\right|\sqrt{\mu }e^{i\frac{\delta ++\delta -}{2}}\rangle \;⊗\;|\sqrt{\mu }e^{\frac{\delta +-\delta -}{2}}\rangle .$$

In Eve’s view, the dual light field state can be regarded as a mixed state represented by $|\varphi \rangle$ due to the random selection of $\delta_{+}$ over the interval [0,2π). Therefore, the SNS protocol confirms that after transmission through the channel, the dual light field state can be expressed as shown in Equation (Equation 9).(9)$$\rho_{AB}=\sum_{k}p_{k}|\psi_{k}\rangle \langle \psi_{k}|.$$

In Equation (Equation 9), $|\psi_{k}\rangle$ denotes the component of the joint state of the two optical fields containing a total of *k* photons, while $p_{k}$ represents the probability of this joint state occurring. This indicates that the decoy-state method can be applied to estimate the count rate and bit error rate of the single-photon component based on the X-data. Specifically, this applies to partial signals corresponding to zero-, single-, and two-photon components, where k = 0, 1, or 2 can be expressed as the following equations.(10)$$|\psi_{0}{\rangle =|0\rangle }_{A}{|0\rangle }_{B},p_{0}=e^{-2\mu },$$(11)$$|\psi_{1}\rangle =\frac{1}{\sqrt{2}}{(|0\rangle }_{A}{|1\rangle }_{B}+e^{i\Delta }{|1\rangle }_{A}{|0\rangle }_{B}),p_{1}=2\mu e^{-2\mu },$$(12)$$|\psi_{2}\rangle =\frac{1}{\sqrt{2}}{(|0\rangle }_{A}{|2\rangle }_{B}+2e^{i\Delta }{|1\rangle }_{A}{|1\rangle }_{B}+e^{i2\Delta }{|2\rangle }_{A}{|0\rangle }_{B}),p_{2}=2\mu^{2}e^{-2\mu },$$(13)$$\Delta =\delta_{A}+\gamma_{A}-\delta_{B}-\gamma_{B}.$$

Under untrusted source conditions, as opposed to ideal light source conditions, each communicating party emits arbitrary quantum states as(14)$$|\psi_{A(B)}\rangle =\sum_{n=0}^{\infty }e^{in(\delta_{A(B)}+\gamma_{A(B)})}\sqrt{P_{n,A(B)}\left(\mu \right)}{|n\rangle }_{A(B)}.$$

Referring to the SNS protocol [25], a new variable $\delta_{\pm }=\delta_{A}\pm \delta_{B}$ is defined, which leads to Eve’s observation that the joint state sent by the two communicating parties can be expressed as(15)$$\rho_{AB}=\int_{0}^{2\pi }P\left(|\psi_{A}\rangle ⊗|\psi_{B}\rangle \right)d\delta_{+},$$$P\left(|x\rangle \right)=|x\rangle \langle x|$ refers to the density operator. Calculating and normalizing Equation (Equation 15) yields the result.(16)$$\rho_{AB}=\sum_{n}p_{n}\left(\mu \right)|\psi_{n}\rangle \langle \psi_{n}|,$$(17)$$|\psi_{n}\rangle =\frac{1}{\sqrt{P_{n}\left(\mu \right)}}\sum_{k=0}^{n}\sqrt{P_{k,A}\left(\mu \right)P_{n-k,B}\left(\mu \right)}{|k\rangle }_{A}{|n-k\rangle }_{B},$$(18)$$p_{n}\left(\mu \right)=\sum_{k=0}^{n}P_{k,A}\left(\mu \right)P_{n-k,B}\left(\mu \right).$$

Further analysis shows that when the light source conditions of both communicating parties are consistent,(19)$$P_{k,A}\left(\mu \right)=P_{k,B}\left(\mu \right)=P_{k}\left(\mu \right),$$
the single-photon component of $\rho_{AB}$ can be expressed as(20)$$|\psi_{1}^{\prime }\rangle =\frac{1}{\sqrt{2}}\left({|0\rangle }_{A}{|1\rangle }_{B}+e^{i\Delta }{|1\rangle }_{A}{|0\rangle }_{B}\right).$$

This means that the same results can be obtained as with ideal light sources even under untrusted light source conditions. Therefore, the security analysis method of the original SNS protocol remains valid. Under untrusted source conditions, the security of the protocol can still be guaranteed.

### 2.3. Parameters Estimation with UB

In the original SNS protocol, Alice (Bob) sends a coherent signal [25], where the average number of photons follows a Poisson distribution(21)$$P_{n,A}\left(\mu \right)=P_{n,B}\left(\mu \right)=P_{n}\left(\mu \right)=e^{-\mu }\frac{\mu^{n}}{n!},$$
we can then calculate Equations (Equation 5)–(Equation 7) using Equation (Equation 21). However, under untrusted light conditions, the photon distribution no longer follows a Poisson distribution. Therefore, light source monitoring is required to estimate $p_{0}^{L(U)}\left(\mu_{i}\right)$, $p_{1}^{L(U)}\left(\mu_{i}\right)$ and $p_{2}^{L(U)}\left(\mu_{i}\right)$.

The light structure of the SNS protocol is similar to that of the MDI protocol with both protocols sharing similarities in phase and intensity modulation. The details of the SNS protocol, which is based on the UB monitoring scheme, are shown in Figure 1. The communicating parties estimate the probabilities of zero-, one-, and two-photon signals $P_{k,A}\left(\mu \right),P_{k,B}\left(\mu \right)\left(k=0,1,2\right)$. Specifically, the monitoring parameter can be obtained through the UB light monitoring structure, which is the UB signal ratio, and it is denoted as Δ. Δ is expressed by the following inequality:(22)$$1-\Delta \leq \sum_{N=N_{\min}}^{N_{\max}}P\left(N\right)\leq 1.$$

The UB signal is defined as photons $n_{a}$ and $n_{b}$, which are prepared by the communicating parties and located in the signal intervals $\left[N_{\min}^{A},N_{\max}^{A}\right]$ and $\left[N_{\min}^{B},N_{\max}^{B}\right]$, respectively. 1−Δ represents the probability that both communicating parties send UB signals. By applying Equation (Equation 22), the probabilities of zero-, single-, and two-photon signals in the secure bit rate formula can be further estimated using the following key parameters(23)$$P_{i}^{\alpha }=\sum_{N=0}^{\infty }P\left(N\right)P_{N}^{\alpha }\left(i\right),\left(i\in \left\{0,1,2\right\}\right).$$

$P_{N}^{\alpha }\left(m\right)$ is the probability that the final signal after transmission contains *m* photons when the original signal has *N* photons with intensity α.(24)$$P_{N}^{\alpha }\left(m\right)=C_{N}^{m}\eta_{\alpha }^{m}{\left(1-\eta_{\alpha }\right)}^{N-m}.$$

As a result, with the analysis shown in Appendix A, we can obtain the upper and lower bounds of $P_{i}^{\alpha }$.(25)$$P_{0}^{\alpha }\geq \left(1-\Delta \right){\left(1-\eta_{\alpha }\right)}^{N_{\max}},$$(26)$$P_{0}^{\alpha }\leq \Delta +\left(1-\Delta \right){\left(1-\eta_{\alpha }\right)}^{N_{\min}}+\Delta {\left(1-\eta_{\alpha }\right)}^{N_{\max}+1},$$(27)$$P_{1}^{\alpha }\geq \sum_{N=N_{\min}}^{N_{\max}}P\left(N\right)N_{\min}\eta_{\alpha }{\left(1-\eta_{\alpha }\right)}^{N_{\min}-1}\geq \left(1-\Delta \right)N_{\min}\eta_{\alpha }{\left(1-\eta_{\alpha }\right)}^{N_{\min}-1},$$(28)$$\begin{array}{cc} P_{1}^{\alpha } & \leq \Delta \left(N_{\min}-1\right)\eta_{\alpha }{\left(1-\eta_{\alpha }\right)}^{N_{\min}-2}+\left(1-\Delta \right)N_{\max}\eta_{\alpha }{\left(1-\eta_{\alpha }\right)}^{N_{\max}-1}\\ \; & +\Delta \tilde{N}\eta_{\alpha }{\left(1-\eta_{\alpha }\right)}^{\tilde{N}-1}, \end{array}$$(29)$$\begin{array}{c} P_{2}^{\alpha }\geq \left(1-\Delta \right)\frac{N_{\min}\left(N_{\min}-1\right)}{2}\eta_{\alpha }^{2}{\left(1-\eta_{\alpha }\right)}^{N_{\min}-2}, \end{array}$$(30)$$\begin{array}{c} F_{2}^{\alpha }\leq \Delta C_{N_{\min-1}}^{2}\eta_{\alpha }^{2}{\left(1-\eta_{\alpha }\right)}^{N_{\min}-3}+\left(1-\Delta \right)C_{N_{\max}}^{2}\eta_{\alpha }^{2}{\left(1-\eta_{\alpha }\right)}^{N_{\max}-2}+\Delta C_{\bar{N}}^{2}\eta_{\alpha }^{2}{\left(1-\eta_{\alpha }\right)}^{\bar{N}-2}. \end{array}$$The previous analysis precisely determines the upper and lower bounds of $P_{i}^{\alpha }$. Furthermore, by substituting the results into Equations (Equation 3) and (Equation 4), the key parameters—the single photon response rate and bit error rate—can be estimated for the protocol’s security analysis. Finally, the secure bit rate *R* is calculated using the formula in Equation (Equation 2).

## 3. Performance with Numerical Simulation

The key parameter in the secure bit rate formula for the UB light monitoring scheme is the UB signal proportion 1−Δ, which is determined by the actual light source signal characteristics and the UB signal interval $\left[N_{\min},N_{\max}\right]$. Δ can be expressed as the following equation [45](31)$$\Delta =G\left(\mu \right)=1-\frac{1}{2}\left[\mathrm{erf}\left(\frac{N_{\max}-\mu }{\sqrt{2\mu }}\right)-\mathrm{erf}\left(\frac{N_{\min}-\mu }{\sqrt{2\mu }}\right)\right],$$
when the light source is considered ideal, $u=\bar{N}$ and the Δ can be directly calculated.

However, in practical analysis, the laser device is affected by the working environment, and fluctuations in light intensity are commonly observed. To simulate the experimental environment, we introduce the light intensity fluctuation parameter $\sigma_{0}$ and set Δ as(32)$$\Delta =1-\int_{N_{min}}^{N_{max}}\left[1-G\left(\mu \right)\right]P\left(\mu \right)d\mu ,$$(33)$$P\left(\mu \right)=\frac{1}{\sqrt{2\pi }\sigma }\exp\left(-\frac{{\left(\mu -\mu_{0}\right)}^{2}}{2\sigma^{2}}\right),$$(34)$$\sigma =\sigma_{0}\times \bar{N},\mu_{0}=\bar{N}.$$

Other calculation parameters are similar to those in the original SNS [25,26] protocol. To accurately estimate the performance of the UB light monitoring scheme, it is necessary to select an appropriate light source signal range based on experimental conditions to determine the UB signal proportion, which is followed by the calculation of the ideal secure bit rate. To simulate the experimental environment, we select the average photon number $\bar{N}=1\times 10^{7}$, the light intensity fluctuation parameter $\sigma_{0}=1\%$ and the interval parameter δ=0.043. Additionally, the traversal interval is set to $\left[N_{min},N_{max}\right]=\left[\left(1-\delta \right)N,\left(1+\delta \right)N\right]$ followed by performance simulation under these conditions.

Based on the parameter estimation results from the previous section, simulations of the SNS protocol using the UB monitoring scheme can be performed under untrusted source conditions. First, we perform a performance simulation of the SNS protocol using ideal experimental parameters, which are set to be the same as in Reference [25] and listed in Table 1. The simulation results are shown in Figure 2. They demonstrate that the protocol can achieve performance close to that of an ideal light source environment, even under untrusted source conditions, when the UB light source monitoring is introduced. Under the set monitoring conditions, the maximum secure transmission distance can reach 817.5 km, which is over 95% of the ideal light source environment.

Furthermore, we can analyze the performance of this monitoring scheme under untrusted source conditions and actual simulation parameters. The simulation results are shown in Figure 3. To simulate a real-world QKD system, we also consider factors such as a high dark count rate and low detection efficiency. Additionally, factors such as fiber attenuation, inherent system errors, and non-ideal data coordination efficiency are also considered in a non-ideal environment. Finally, the experimental parameters are chosen based on Table 2. The simulation result shows that under untrusted source conditions, the maximum secure transmission of the SNS protocol based on the UB monitoring scheme can reach 490 km, which is over 92% of the ideal light source environment. According to this result, although combining the UB monitoring scheme requires sacrificing some signals outside the UB interval, the performance can still be maintained within an acceptable range under certain experimental conditions.

## 4. Conclusions

Under untrusted source conditions, the primary factors to consider are the security of the dual light-field protocol as well as sufficient secure bit rate and transmission. In this paper, we introduce UB light monitoring into the SNS protocol. The introduction of UB light-source monitoring in the SNS protocol provides a novel and effective solution. By externally monitoring the light source signals of the communicating parties, following filtering and phase randomization, the proportion of UB signals with photon numbers within a specified interval $\left[N_{\min},N_{\max}\right]$ can be determined. This allows for the accurate estimation of key parameters in the protocol’s security bit rate formula, such as the probabilities of zero-photon, single-photon, and two-photon signals. Consequently, a compact lower bound for the security bit rate can be effectively determined. Furthermore, this scheme monitors the laser source signal before attenuation, bypassing the challenges of low detection efficiency and high costs commonly associated with single-photon detection for weak light monitoring.

The simulation analysis demonstrates that this scheme can achieve 817.5 km of secure transmission under ideal simulation parameters, which closely matches the transmission performance of an ideal light source environment. Under actual simulation parameters, the scheme can achieve 490 km of secure transmission, matching over 92% of the performance in an ideal light source environment. Compared to other protocol schemes, this scheme demonstrates better tolerance to light source fluctuations and offers significant advantages in transmission distance and secure bit rate.

## Figures and Tables

**Figure 1 entropy-27-00547-f001:**
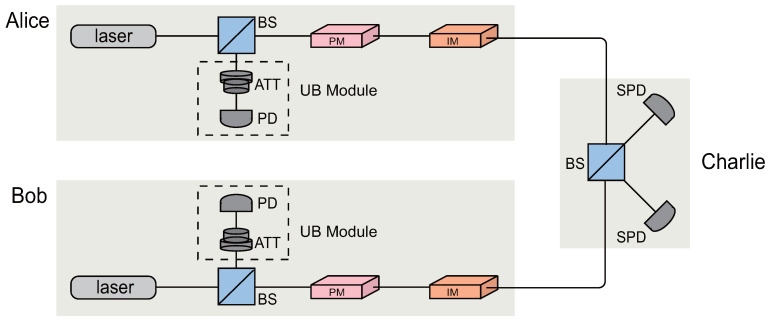
The structure of the sending-or-not-sending (SNS) protocol with an extra UB module in each of Alice’s and Bob’s parts. The UB module is composed of an attenuator (ATT) and a photon detector (PD).

**Figure 2 entropy-27-00547-f002:**
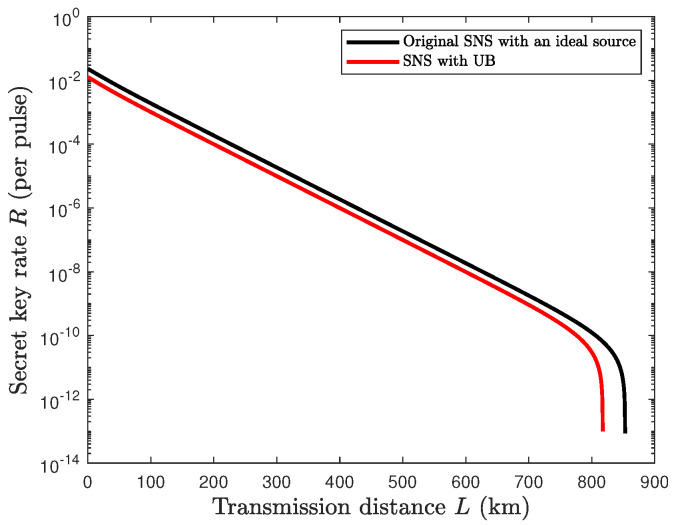
Performance diagram of SNS protocol based on UB monitoring solution under ideal parameters.

**Figure 3 entropy-27-00547-f003:**
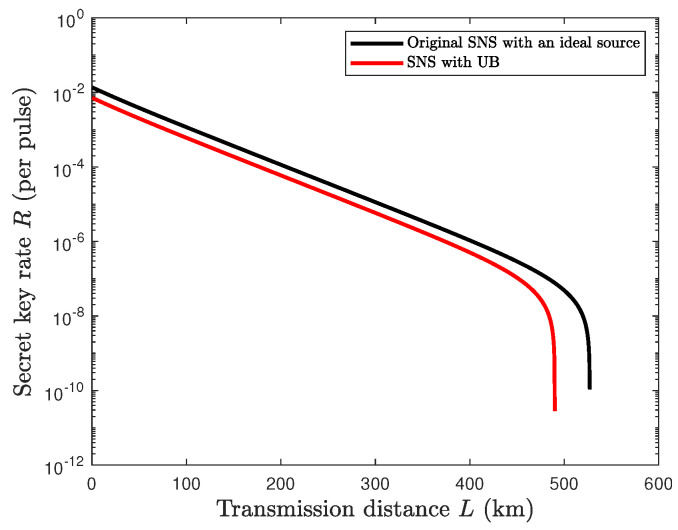
Performance diagram of SNS protocol based on UB monitoring solution under actual parameters.

**Table 1 entropy-27-00547-t001:** Values of parameters used in simulation. α: the fiber loss coefficient (unit: dB/km); $p_{d}$: the dark count rate of the detector; $\eta_{D}$: the detection efficiency; $e_{det}$: the misalignment error of the QKD system; *f*: the error correction efficiency.

α	$p_{d}$	$\eta_{D}$	$e_{det}$	*f*
0.2 dB/km	$1\times 10^{-11}$	0.8	1%	1.1

**Table 2 entropy-27-00547-t002:** Values of parameters used in simulation (set as in Refs. [24,46] for more practical conditions).

α	$p_{d}$	$\eta_{D}$	$e_{det}$	*f*
0.2 dB/km	$1\times 10^{-8}$	0.6	2%	1.15

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
