# Peer review of "An Effective Parameter Analysis for Sending-or-Not-Sending Quantum Key Distribution with Untrusted Light Sources"

_entropy, 2025, doi:10.3390/e27060547_

Round 1

Reviewer 1 Report

Comments and Suggestions for Authors

My comments is attached.

Author Response

Thank you very much for taking the time to review this manuscript. We greatly appreciate your valuable feedback and insightful suggestions. According to your 5 comments, we revised our manuscript as below:

(1). We appreciate the reviewer for pointing out the formatting issues and grammatical errors. We have corrected all of them accordingly.

(2). In our preliminary work, the implementation cost of the former scheme is high, as it requires the addition of extra Single-Photon Detector (SPD). Moreover, the parameters of the SPD, such as detector efficiency, are easily influenced by environmental factors, causing the actual monitoring results to deviate from the theoretical expectations. Furthermore, the former scheme still maintains implicit assumptions about the photon number distribution, failing to address the problem effectively.

(3). The ratio of BS Δ represents the ratio of pulses that are tagged. The photon number of pulses emitted by the laser source in a QKD system is likely to fall within a specific interval , where N represents the average number of pulses. The range of above interval is controlled by . This subset of pulses is defined as untagged pulses (UB signals), while the remaining pulses are considered tagged pulses. Only the tagged pulses are considered in the security analysis. After Alice and Bob determine the probability of a pulse being tagged or untagged, they can estimate the upper and lower bounds of the gain and QBER for untagged pulses. This estimation is based on the relationship between the overall gain and QBER, enabling the calculation of the secure bit rate.

(4). In Fig. 3, the original SNS protocol with an ideal source represents the SKR optimized for light intensity combinations. We traverse the light intensity range to determine the optimal light intensity at each transmission distance.

(5). We have extended the introduction to the references for the high-dimensional entanglement quantum key distribution protocol and added several sentences in lines 22-27 to extend the introduction part.

In general, the revisions to the review comments are listed below:

We have carefully checked all grammatical errors and formatting issues in the re-submitted files. We have added several sentences in lines 22-27 to extend the introduction part, introduced more about the high-dimensional entanglement quantum key distribution protocol. We have explained the meaning of the ratio of BS in Fig.1 and the original SNS protocol with an ideal source in Fig.3. These changes are marked in red in our revised manuscript, and the reviewer can see the attached PDF file for details. We hope that the revision can be accepted by the reviewer.

Thank you again for your comments and suggestions!

Reviewer 2 Report

Comments and Suggestions for Authors

An Parameter Analysis for Sending-or-Not-Sending Quantum
Key Distribution with Untrusted Light Sources 
by J Huang, W Li, Y Qiao

Twin‑field (TF) quantum key distribution, has inspired a family of TF‑type protocols that boost transmission distance and security. The most widely adopted variant, the Sending‑or‑Not‑Sending (SNS) protocol, achieves long‑range security by judiciously choosing when to transmit or withhold signals. Subsequent work has tightened SNS security proofs under varied assumptions extended its reach using independent lasers, actively odd‑parity pairing, and phase‑post‑selection. Despite these advances, practical vulnerabilities—especially those tied to imperfect light sources—remain. Earlier “light‑source monitoring” approaches were largely theoretical. This paper proposes a deployable counter‑measure, Untagged Bits (UB), that mitigates light‑source–related security risks in real‑world SNS implementations.

Twin‑field quantum key distribution (QKD) has unlocked record transmission distances, and the SNS variant is its most practical form, yet real‑world deployments still suffer from security loopholes created by fluctuating, untrusted light sources. According to the authors, this research closes that gap by embedding an Untagged‑Bits (UB) light‑monitoring module directly into the SNS protocol: the module samples the laser before attenuation, applies filtering and phase randomisation, and determines the fraction of pulses whose photon numbers lie in a safe window [Nmin,Nmax]. Those statistics yield tight, protocol‑level estimates for zero‑, single‑ and two‑photon events, enabling a compact, provable lower bound on the secure key rate without costly single‑photon detectors. Simulations show the UB‑enhanced SNS scheme achieves 817 km of secure transmission under ideal parameters and 490 km (> 92 % of ideal performance) under realistic conditions—distances that surpass competing countermeasures while maintaining high key rates and strong tolerance to source noise. In short, the work delivers the first field‑ready solution that preserves near‑ideal SNS performance and security with minimal hardware overhead, pushing practical QKD toward continental‑scale networks.

considering the above I believe this research is worth publishing so that my recommendation is to accept the manuscript in its present form.

Author Response

Thank you very much for your valuable time and insightful suggestions. We are very grateful to hear that you recommend our manuscript for publication in its current form. We appreciate your recognition of our work and the constructive comments you provided during the review process, which significantly helped us improve the quality of our manuscript.

We look forward to the final decision from the editor and are happy to make any further modifications if necessary.

Thank you again for your support and encouragement.

Reviewer 3 Report

Comments and Suggestions for Authors

In the manuscript entropy-3621216, the authors discuss issues related to quantum key distribution protocols, focusing in particular on the security of these protocols in real-world environments. The authors show that solutions based on untagged bits provide good monitoring performance.

The topic of the manuscript is interesting and has potential implications for quantum technology. The results presented in the submitted paper are interesting and valid enough to be published, and the authors provide a sufficient list of publications introducing the reader to the topics discussed. However, in my opinion, the current state of the manuscript is not suitable for publication.

There are many typographical errors in the text. At the end of most sentences, there is no space after the period that ends the sentence. There is also no space in other places - see, for example, page 10, line 254. Also, there should be a comma after equations (1, 12, 17, etc.) and a period after equations (7, 14, etc.). Are the authors sure that in lines 107 and 123 should be Eve instead of Charlie?
In line 24 there repeated 'However'.

In general, authors should read the entire text carefully and correct errors and typos.

Finally, it can be stated that the article can be accepted for publication after the revision along the above mentioned points.

Author Response

Thank you very much for taking the time to review this manuscript. We greatly appreciate your valuable feedback and insightful suggestions. According to your comments, we revised our manuscript as below:

1.We appreciate the reviewer for pointing out the formatting issues and grammatical errors. We have corrected the improper formatting accordingly.

2.In lines 107 and 123 (now updated to lines 114 and 130), we use Eve instead of Charlie because, in the protocol design, Charlie represents the third-party detector. In the security analysis, to investigate eavesdropping behavior, the third-party detector and potential eavesdroppers are collectively referred to as Eve. We describe Eve's observations when she attempts to eavesdrop on the information. After deriving the formulas, we obtain the same results as those in the original SNS protocol. Therefore, we apply the security analysis method of the original SNS protocol to ensure the security of SNS with UB.

In general, the revisions to the review comments are listed below:

We have carefully checked all grammatical errors and formatting issues in the re-submitted files. We have explained the reason for using Eve instead of Charlie. These changes are marked in red in our revised manuscript, and the reviewer can see the attached PDF file for details. We hope that the revision can be accepted by the reviewer.

Thank you again for your comments and suggestions!